# The Use of Drones to Determine Rodent Location and Damage in Agricultural Crops

Dor Keshet [1,2], Anna Brook [3], Dan Malkinson [2,3], Ido Izhaki [1] and Motti Charter [2,3,*]

1 Department of Evolutionary and Environmental Biology, University of Haifa, Mount Carmel, Haifa 3498838, Israel

2 Shamir Research Institute, Katzrin 1290000, Israel

3 Department of Geography and Environmental Studies, University of Haifa, Mount Carmel, Haifa 3498838, Israel

* Correspondence: mcharter@geo.haifa.ac.il; Tel.: +972-544901130

**Abstract:** Rodent pests cause extensive damage to agricultural crops worldwide. Farmers' ability to monitor rodent activity and damage within crops is limited due to their inability to simultaneously survey vast agricultural areas for rodent activity, the inability to enter certain fields, and the difficulty of monitoring rodent numbers, as well as using traps due to trap shyness and high labor costs. Drones can potentially be used to monitor rodent numbers and damage because they can cover large areas quickly without damaging crops and carry sensors that provide high-resolution imagery. Here, we investigated whether rodent activity (Levant voles *Microtus guentheri* and house mice *Mus musculus*) is related to vegetation health and biomass in Alfalfa (*Medicago sativa*) fields. We used a drone to photograph one hundred and twenty $10 \times 10$ m plots in nine fields and calculate the plots' normalized difference vegetation index (NDVI) and biomass. On each plot, we also trapped rodents, counted rodent burrows, and evaluated the harvested dry crop yield. The number of burrows was positively related to the number of Levant voles trapped ($F_{1,110} = 12.08$, $p < 0.01$) and negatively related to the number of house mice trapped ($F_{1,110} = 5.23$, $p < 0.05$). Biomass extracted from drone images was positively related to the yield harvested by hand ($F_{1,83} = 3.81$, $p < 0.05$). Farmers, therefore, can use burrow counting in place of trapping Levant voles, and biomass estimates from drones can be used in place of manual yield calculations. NDVI ($F_{1,95} = 73.14$, $p < 0.001$) and biomass ($F_{1,95} = 79.58$, $p < 0.001$) were negatively related to the number of Levant voles trapped, and the number of burrows were not related to the number of house mice trapped. We demonstrate that drones can be used to assist farmers in determining the Levant vole presence and damage within crop fields to control rodents using precision agriculture methods, such as adding rodenticides in specific areas, thus increasing efficiency and decreasing the amount of pesticides used.

**Keywords:** drone; remote sensing; rodent damage; Alfalfa; NDVI; biomass

## 1. Introduction

The crop yield depends on the climate, nature of the soil, field topography, amount of agricultural inputs (fertilizers, irrigation water, etc.) applied, patterns of irrigation and fertilizer practices [1–3], and extent of damage due to rodents [4–6]. Therefore, controlling rodent pests is essential to ensure food security worldwide [7], especially since global crop demand is projected to increase by 70% by 2050 [8]. Currently, rodent trapping is the most common method to determine rodent numbers and densities. However, for certain crops, farmers cannot access fields because this would cause additional crop damage [9,10]. Additionally, trapping rodents has high labor and material costs. Still, in crops, trapping can be used, but it is labor-intensive and expensive due to the high initial costs of purchasing traps and ongoing labor costs, and it requires sustained efforts to be effective. On the other hand, chemical rodenticides can be highly effective at controlling specific rodent pest

populations [11] but are costly to use and become less efficient over time as rodents become resistant and bait-shy [12]. Further, there are worldwide environmental concerns, as some rodenticides can cause secondary poisoning of non-target wildlife species [13].

Vole species (genus *Microtus*) are known as significant agricultural pests worldwide [14,15]. For example, in Israel, Levant voles (*Microtus guentheri*) are significant pests in field crops [16], where vole populations can accumulate thousands of burrow openings per hectare [17] due to their very high reproductive output. Voles reach reproductive age within the first month of birth, their gestation period is only 21 days, and they raise five to six young per litter [18,19]. As a result, vole populations fluctuate, and when they erupt, they cause extensive agricultural loss [20].

In addition to Levant voles, House mice (*Mus musculus*) are also common in alfalfa fields in Israel [21]. Still, unlike voles, mice most likely do not create large burrowing infrastructures, and their burrows may be located on the border of the fields and not inside the field. Even though house mice have been suggested as agricultural pests [5,16], in Israel, voles are thought to be the leading cause of crop damage [22]. Therefore, there is a need to determine whether voles and mice are related to burrows and damage in the field.

One major challenge when controlling rodent numbers is that farmers need an efficient means to determine the distribution and abundance of rodents in a field. Typically, farmers begin noticing the presence of rodents when the population is already large, when there are many burrows, and the damage is so intense that bald sections within the field are visible to the eye. A means to determine the location of rodents when the population number is small might allow farmers to exert control in specific areas more intensively before the vole population grows too large. Unfortunately, not only do rodents decrease crop yield, but the rodenticides used to control them are expensive (due to both purchasing and distribution costs) and harmful to the environment.

On one hand, farmers are under intense pressure to reduce rodenticide use; on the other hand, they need to grow more crops with fewer resources, such as rodenticides, due to regulations. Precision agriculture is a process that involves the collection, interpretation, planning, and/or use of data about a particular field and crop [2] and could assist in reducing vole numbers. Vole populations are found in 'hot spots', patches with many voles that reproduce and inflict crop damage in the immediate vicinity around their holes. Since voles cause damage to crops in specific areas within fields, it is first necessary to determine where the voles are to concentrate the control of voles in these particular areas.

The application of remote sensing methods to identify rodents directly is challenging because of their nocturnal cryptic behavior and size. Although it may be possible in the future [23], an alternative approach may be to identify rodent burrows and damage to vegetation [24], which could serve as proxies for rodent presence and damage. Rodent activity has been studied using satellite imagery by identifying rodent species with vast burrowing complexes [25,26]. It is not possible to determine the within-field variation of rodent activity using satellite images due to the low resolution of such images and the small size of most rodent burrows. Drones may provide a feasible alternative.

Small drone platforms have been used in both precision agriculture and wildlife surveys [27]. The benefits of drones over manual and other survey methods are that it is possible to control the number of re-visits to the same area [28] and collect finer spatial resolution data on both temporal and spatial scales [29]. Compared to satellite images, which provide more extensive area cover at a lower spatial resolution, or manned aircraft, which cover large areas at a high resolution but are expensive, drones can provide very high-resolution data on agricultural fields at a lower cost [27]. Rodent burrows have been identified from drone images using object-oriented supervised classification [30] and machine learning [31].

Precision agriculture multispectral imaging is a widely used method, primarily due to the correspondence between plant respiratory processes and absorption features across the near infrared range (800 nm to 1000 nm NIR)of the electromagnetic spectrum placed in particular on the red edge (680 nm to 730 nm) [32] and their correlation with plant metabolic

processes [33]. For example, a promising study in Spain found that the normalized difference vegetation index (NDVI) was related to the number of common vole (*Microtus arvalis*) burrows [24] and suggested using this index to assess and monitor vole damage.

Even though the existing drone systems can capture these spectral bands (e.g., Micasense Rededge), they are expensive, have a lower spatial resolution, and are more complicated to operate. The digital (RGB) cameras across the visible range of electromagnetic spectrum on drones are small, low-cost, and user-friendly. Therefore, developing the ability to use RGB cameras on drones that will provide similar information would be highly beneficial due to lower costs, simplicity, and rapid field-scale operation.

Neural networks have been proposed [34] to generate NIR bands from RGB data using a nonlinear transformation and mapping from RGB to NIR. It is a great challenge, yet several recent studies showed successful and promising results [35–37]. Additional methods [34] performed by mapping between RGB values and high dimensional spectral signature, which is not a trivial problem, aimed to produce spectral recovery. Examples include radial basis function network [38], K-SVD based sparse coding [39], nonnegative sparse coding [40], and manifold-based models [41], highlighting that properly designed nonlinear mapping can significantly increase recovery accuracy.

Before drones can replace traditional methods to determine rodent abundance in fields, we first need to determine whether vegetation indexes measured by drones at a low altitude are related to rodent numbers. Specifically, the objective of the current study was to examine whether Levant voles and house mice densities are related to NDVI and biomass measured by drones in Alfalfa fields, as estimated by burrow counts and trapping. We hypothesized that areas with more rodents would have lower NDVI values and biomass due to damage and stress caused by rodents to crops. We further aimed to determine whether rodent burrows could be used as an index of Levant vole activity and whether biomass calculated from remote sensing can be used as a proxy for crop yield.

## 2. Materials and Methods

### 2.1. Study Area

The study was conducted in nine Alfalfa fields located in the Hula Valley, Israel (33°6′15.77″ N, 35°36′26.11″ E) from 6 February to 30 May 2018. None of the fields were treated for rodent management (no rodenticides), and all the fields had full-grown alfalfa of around 50 cm in height. All the flights were conducted from 10:30 AM to 11:45 AM to avoid the negative effect of the shadows. Each field had 8–24 plots (N = 112 plots) of 100 m$^2$ (10 × 10 m) sampling plots that were selected randomly, located = >20 m from the edge of the fields and = >100 m apart from each other. Therefore, the number of plots varied between the size of the fields. The size of plots was selected because rodent densities can be patchy, as found in a previous study that showed the number of rodents and vegetation indexes can fluctuate greatly between 100 m$^2$ plots (Charter unpubl. data). The corner of each plot was marked with a red flag, and GPS coordinates were recorded using a Leica GS18 GNSS RTK system in the field. Each sampling plot was photographed using a drone. Additionally, in each plot, rodent burrows were counted, their cover was estimated, and crop yield was sampled. Plots were photographed before entering to trap the rodents and count the rodent burrows to avoid damaging the vegetation.

### 2.2. Rodent Trapping

Twenty rodent snap traps (7.6 × 13.9 cm, Kness Manufacturing, Albia, IA, USA) without bait were placed perpendicular to rodent runways between the burrows, on flat ground, and were not covered by alfalfa within three days of when the fields were photographed. In a previously unpublished study, we found that traps not baited captured both Levant voles and house mice, whereas baited traps only captured the latter (Charter unpubl. data). The traps were added in the evening and collected early the following morning to prevent catching diurnal animals. Rodents were not trapped during a full moon since rodent activity decreased [42]. We also compared the number of trapped voles [18] to



vole burrow counts [43]. The trapping was conducted under permit number 2018/41815, provided by the Israel Nature and Parks Authority.

### 2.3. Rodent Burrows

Even though rodent burrows have been found to be related to common voles [43] and are used as an index of vole density because of the ease of counting burrows compared to trapping, it is still unknown whether the same relationship exists in Levant voles in Israel [21,44,45]. Each sampling plot was divided into five strips of 1 × 10 m, and all rodent burrows were counted by one of the authors (DK) after the rodent traps were collected. Even though vole burrows can be counted using an object detection model [31] when the vegetation is low, in this study, the alfalfa was high and covered the visibility of the burrows, making it impossible to count using drone images.

### 2.4. Crop Biomass and Yield Sampling

After photographing, rodent trapping and counting of burrows, we hand-harvested the vegetation by placing two 1-m$^2$ metal frames and trimming the alfalfa to the very base in each plot. To estimate dry biomass, the crop was weighed and dried for 48 h at 105 °C in an oven and weighed again after drying.

### 2.5. Drone-Based Remote Sensing

We first flew a DJI Phantom 4 pro drone equipped with an inbuilt three-axis gimbal-stabilized DJI FC6310 20MP RGB camera (DJI, Shenzhen, China). The position and orientation parameters for the camera were provided by the onboard inertial measurement unit (IMU) and global positioning system (GPS). To ensure a ground sampling distance (GSD) of 0.5 cm/pixel, the drone was flown at an altitude of 20 m. The GSD was calculated according to (1) imagery collected at the height of 20 m above the ground, (2) the optimized sensor width of the Phantom 4 Pro camera (12.83 mm), (3) the focal length of the camera (24 mm), and (4) image width of 5472 pixels and height of 3648 pixels. The Sentra Field Agent-Drone Mapping application Software (Sentra, Inc., Minneapolis, MN, USA) was used to control the flight and capture the images.

### 2.6. Data Analyses

A mosaic was created for each plot using Arcmap V10.4 software (Esri, Redlands, CA, USA) (Figure 1a). In drone-based imaging, reflectance measurement is influenced by light scattering, directional reflectance, and topography. We applied the Supervised Vicarious Calibration (SVC) approach and ground calibration panels [46] to convert raw RGB pixels to radiance and then to reflectance images. The spectrum of each calibration panel was measured with a field spectrometer (USB4000 OceanOptics, Orlando, FL, USA) in situ in radiance and reflectance modes (calibrated against a 100% white spectral panel).

At the radiance level, the data were processed by a multispectral domain translation generation adversarial model GAN [47], including an independent encoder, decoder, and discriminator for each spectral band. We aimed to estimate the conditional distribution between RGB radiance and NIR (780 nm with a bandwidth of 40 nm) radiance with a learned deterministic mapping function and the joint distribution. To translate from RGB to NIR, a fully shared latent space assumption [48] was adapted, and the encoder obtained the joint probability distribution. In the process of image reconstruction from the RGB imagery to target RGB + NIR imagery, the source encoder is selected from the encoder library, and the target decoder is chosen from the decoder library. The input RGB bands share latent code, and the decoder reconstructs the target NIR image from the latent code. Then, the loss is calculated by the target discriminator. Each pixel is generated from a latent code that is shared by all the data and synthesized by the decoder. The proposed network is inspired by the previously published network [49]. The encoder consists of a set of stride–downsampling convolutional layers and one residual block. The decoder

processes the latent code by a residual block and then restores the image size through a 1/2 strided upsampling convolutional layer.

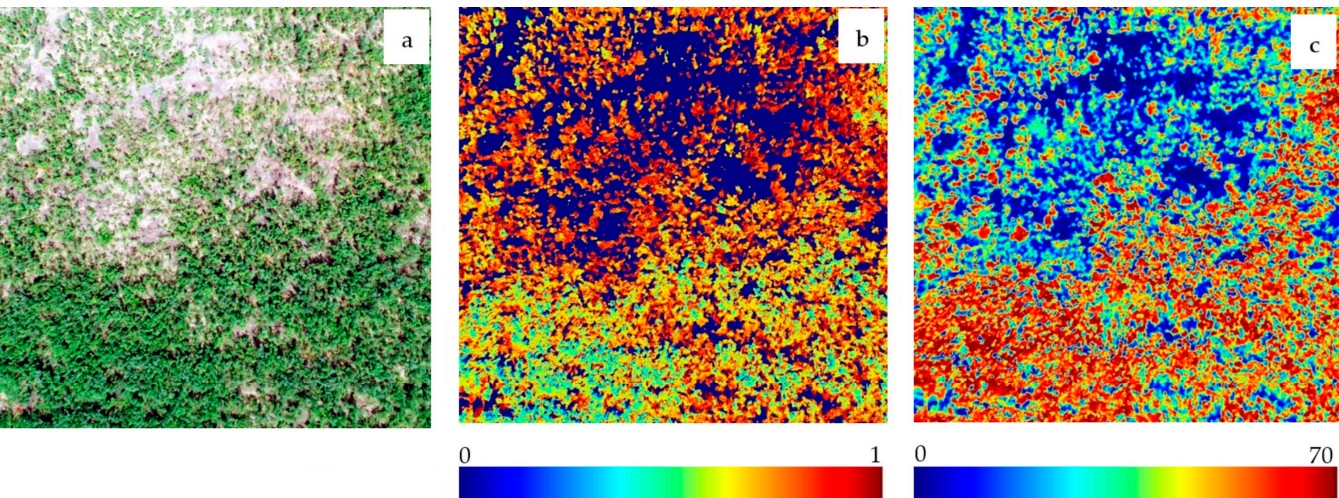

**Figure 1.** An example image of a 100 m$^2$ (10 × 10 m) sampling plot: RGB photo (**a**), NDVI (0 = no vegetation to healthy vegetation) (**b**), and biomass (0 = no biomass to 70 mg biomass per gram of dry matter) (**c**). Overall, the study took place in 9 alfalfa fields; each field had 8–24 plots (N = 112 plots) located =>100 m apart.

The model was trained with RGB patched images and the NIR spectral signature from calibration SVC targets, single crop, and soil predefined plots and tested with images from the field. A predefined uniform 20 × 20 pixel patch of the RGB image was used as input for the model and was translated to NIR spectra collected from the same patch. The input radiance data was augmented by randomly rotating and horizontally flipping the input. The Adam optimizer with default parameters was employed for the network's hyperparameters. The learning rate was set at 0.0001 and decays linearly. The model was trained for 1000 epochs. The inference stage compared the performance of the trained model with a test set from the same dataset, but with larger patches of size 256 × 256.

The patch discriminator predicts the probability value for the input image. We adopt the objective loss function to make the encoder output the same latent space; therefore, the RGB + NIR image can be reconstructed from the latent space. The use of cross-domain reconstruction limits the spatial variation of the encoding and decoding, and it stabilizes spectral translation results.

Since we are using the SVC approach with multiple panels, the spatial domain at this stage might be neglected, as in previously published studies [50]. The total loss function is the weighted sum of the NIR band's counter loss, reconstruction loss, and latent matching loss.

To evaluate the performance of the GAN model, we employed two standard measures, including mean-square-error (MSE) and spectral angle mapping (SAM), to quantify the spectral similarity between the generated and reference NIR data. The reference NIR data were generated from the field spectrometer in situ radiance spectra collected by USB4000 OceanOptics (calibrated against a 100% white spectral panel) over calibration panels and validation targets (selected crops and bare soil plot) and resampled according to the NIR band configuration (780 nm with a bandwidth of 40 nm).

The achieved SAM values were less than 0.05 radian, which is evidence of extensive overlap between the reconstructed and target RGB + NIR spectral data. The MSE values were relatively low, as the radiance data were measured from 0 to 10,000 mW. These results assure accurate translation from the RGB to the NIR spectrum domain and well reconstructed NIR data.

Each reconstructed RGB + NIR mosaic of the plots was spectrally analyzed using ENVI classic V5.5 software (Harris Geospatial Solutions, Broomfield, CO, USA), and two vegetation indices were calculated: the NDVI (Normalized Difference Vegetation Index) (Figure 1b) and the biomass index (Figure 1c). The NDVI is calculated as follows: NDVI = (NIR − RED)/(NIR + RED). The biomass index was calculated as an excess green index outlining a plant region and is computed as follows: ExG = 2 × GREEN − RED − BLUE, where each pixel in a normal RGB image receives a value for the biomass index ranging from 0 to 70 mg biomass per gram of dry matter [51].

The test dataset included 45 drone-based remote-sensing patched images. The average values of the evaluation metrics were calculated (Table 1). Moreover, the generated NIR band was evaluated based on mean absolute error (MAE), and mean absolute percentage error (MAPE) was reported in Table 2. We also assessed the MAE of the resulting NDVI and biomass (Table 2).

**Table 1.** Mean-square-error (MSE) and spectral angle mapping (SAM) values for the test dataset of 45 drone-based remote-sensing patched images.

| | MSE (mW/sq m/Str/nm) | SAM (Angle in Radians) |
|---|---|---|
| SVC * black 100% | 52 | 0.01 |
| SVC grey 50% | 34 | 0.004 |
| SVC grey 25% | 22 | 0.006 |
| SVC grey 17% | 18 | 0.009 |
| SVC white 100% (natural gravel) | 42 | 0.016 |
| Crop | 64 | 0.03 |
| Soil | 24 | 0.007 |

* SVC = Supervised Vicarious Calibration.

**Table 2.** The results reported as mean absolute error (MAE) and mean absolute percentage error (MAPE) of the proposed generation adversarial model (GAN) method, as well as MAE for normalized difference vegetation index (NDVI) and biomass on generated near-infrared (NIR) band.

| | MAE ($10^{-3}$) | MAPE (%) |
|---|---|---|
| GAN model | 18.27 | 1.83 |
| NDVI | 19.45 | |
| Biomass | 22.83 | |

### 2.7. Statistical Processing

We used generalized linear mixed models after testing the data for normality with a log link function to determine whether there was a relationship between rodent numbers, rodent burrows, NDVI, biomass, and yield with field ID as a random effect to avoid pseudo-replication and to control for effect between fields. Significant results were presented throughout, as $p < 0.05$, $p < 0.01$, or $p < 0.001$. All statistical analyses were performed using SPSS V.23 software (IBM Corporation, Armonk, NY, USA).

## 3. Results

### 3.1. Rodent Trapping

During the study, there were 14 nights of trapping, in which 2240 traps were placed, and 291 small mammals were captured: 159 Levant voles, 130 house mice, 1 Tristram's jird (*Meriones tristrami*), and 1 lesser white tooth shrew (*Crocidura suaveolens*). Overall, there were, on average, 1.6 Levant voles trapped (N = 130, SD = 1.9, range = 0–8) and 1.2 mice trapped (N = 99, SD = 1.7, range = 0–7) per plot. The number of Levant voles and house mice trapped were inversely related (generalized linear mixed model with a log link function and with the field as a random variable explaining 6.10% of the random effects R2 variation, $F_{1,110} = 3.62$, $p < 0.05$, Figure 2).

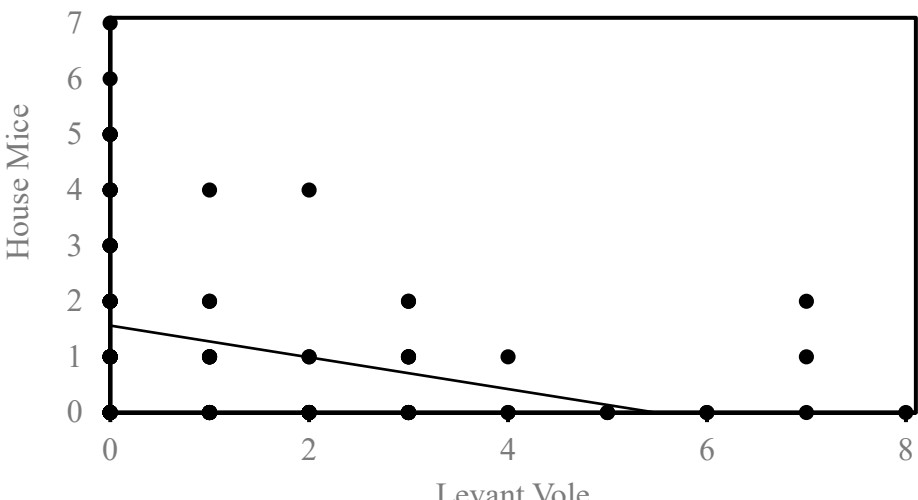

**Figure 2.** Relationship between the number of house mice and Levant voles captured in 112 plots (Statistics in Section 3.1).

*3.2. Relationship between the Number of Rodent Burrows and the Number of Rodents Trapped*

An overall average of 21.5 burrows were counted (N = 1721, SD = 26.5, range = 0–122) per plot. The number of rodent burrows was positively related to the number of Levant voles trapped (generalized linear mixed model with a log link function and with field as a random variable explaining 47.73% of the random effects R2 variation, $F_{1,110} = 12.08$, $p < 0.01$, Figure 3a) and negatively related to the number of house mice trapped (generalized linear mixed model with a log link function and with field as a random variable explaining 48.18% of the random effects R2 variation, $F_{1,110} = 5.23$, $p < 0.05$, Figure 3b). The number of rodent burrows can, therefore, be used as an index of the number of Levant voles.

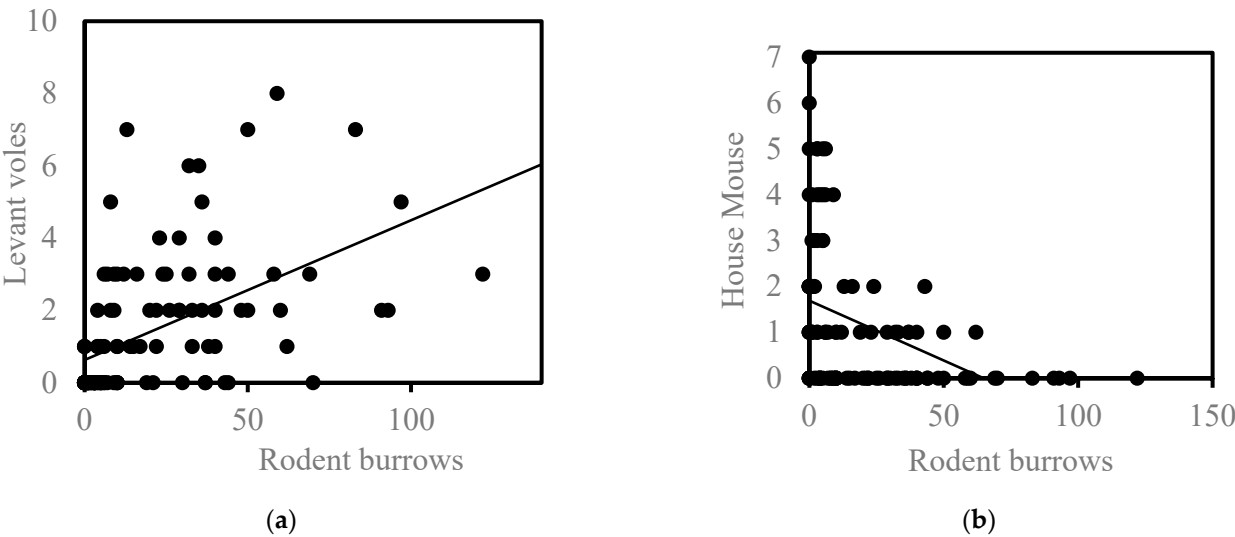

(**a**)                                                                  (**b**)

**Figure 3.** (**a**,**b**) Relationship between rodent burrows and the number of Levant voles and house mice trapped (Statistics in Section 3.2).

*3.3. Relationship between Crop Yield and Biomass*

The manually harvested crop yield was positively related to the biomass estimated using the index proposed above (generalized linear mixed model with a log link function and with field as a random variable explaining 53.56% of the random effects R2 variation, $F_{1,83} = 3.81$, $p < 0.05$, Figure 4).

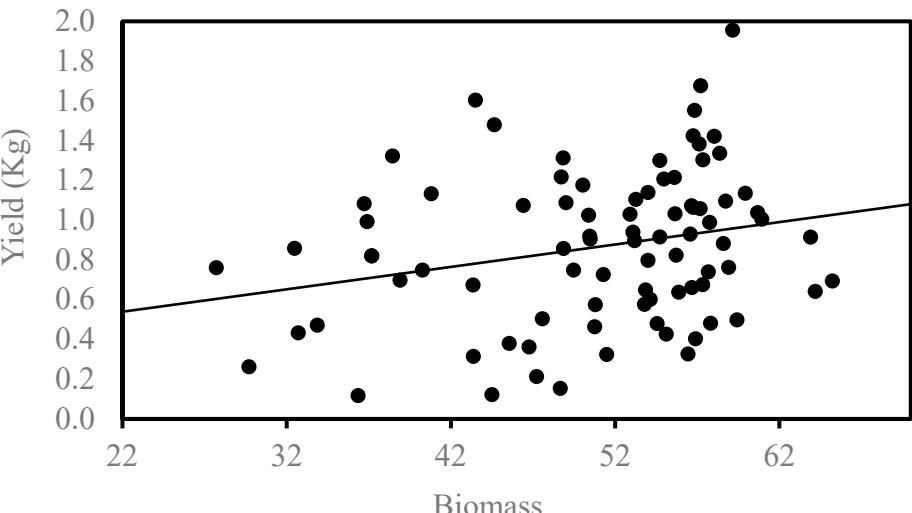

**Figure 4.** Relationship between the yield of alfalfa harvested by hand and the biomass index from the drone images (Statistics in Section 3.3).

*3.4. Relationship of Rodent Burrow Number to NDVI and Biomass*

The number of burrows was negatively and significantly correlated with NDVI values (generalized linear mixed model with a log link function and with field as a random variable explaining 55.55% of the random effects R2 variation, $F_{1,95} = 73.14$, $p < 0.001$, Figure 5a). Similarly, the number of burrows and biomass were negatively and significantly related (generalized linear mixed model with a log link function and field as a random variable explained 59.27% of the random effects R2 variation, $F_{1,95} = 79.58$, $p < 0.001$, Figure 5b).

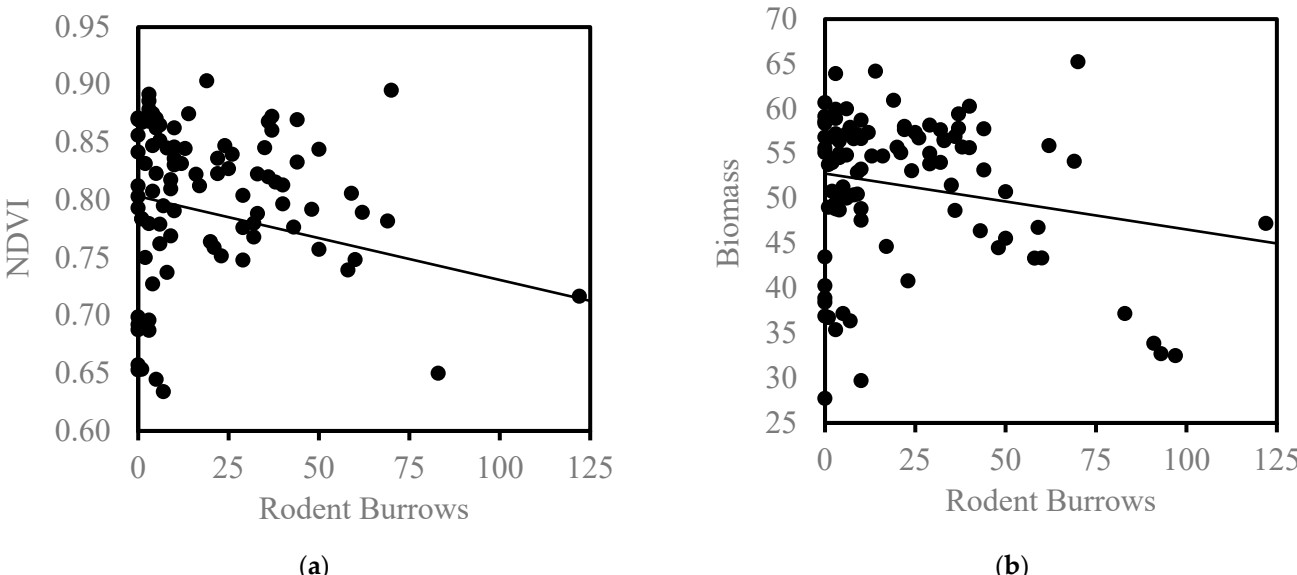

(**a**)                                                                        (**b**)

**Figure 5.** (**a**,**b**) Relationship of rodent burrows to NDVI and biomass (statistics in Section 3.4).

NDVI (generalized linear mixed model with a log link function and with field as a random variable explaining 37.5% of the random effects R2 variation, $F_{1,95} = 4.26$, $p < 0.05$) and biomass (generalized linear mixed model with a log link function and with field as a random variable explaining 46.55% of the random effects R2 variation, $F_{1,95} = 6.66$, $p < 0.01$) were both negatively related to the number of Levant voles trapped, but were not related to the number of house mice trapped (NDVI: generalized linear mixed model with a log link function and with field as a random variable explaining 37.50% of the random effects R2 variation, $F_{1,95} = 0.62$, $p = 0.43$; biomass: generalized linear mixed model with a log link

function and with field as a random variable explaining 44.14% of the random effects R2 variation, $F_{1,95} = 0.33$, $p = 0.57$).

## 4. Discussion

In the current study, drone imagery was used to calculate spectral vegetation indices, which provided detailed, high-spatial-resolution estimates on the health (NDVI) and amount (biomass) of alfalfa. By using burrow counts and rodent trapping, we were able to determine that UAV images can be a very effective tool to identify rodent hotspots and assess rodent damage; however, they cannot be used effectively for all species. We found that Levant vole numbers were negatively related to NDVI and biomass. In contrast, the number of house mice was negatively associated with the number of burrows, but was not related to NDVI and biomass. It is, therefore, important not only to study the effectiveness of using drones within an area, but also to recognize that rodent activity and crop damage may vary among species and potentially among areas. Levant voles, but not house mice, were also the leading cause of crop damage (lower biomass), which has been noted previously [52–54]. This is the first study that has shown a direct relation between voles and crop health. Additionally, since the number of voles and mice were inversely related, future studies are needed to determine whether the species compete.

The NDVI approach was also applied in Spain to determine common vole activity in fields [24]. Even though the design of the Spanish study was different, the pattern of finding lower NDVI in areas with more common voles is similar to that found in Levant voles in Israel. The relationship between NDVI and two species of voles in two different countries using drones is promising. The use of NDVI as an index of rodent activity can be of great assistance to farmers.

Currently, traditional methods of rodent trapping and counting rodent burrows are impossible due to the inability to enter the alfalfa fields without damaging the crops. In this study, we took drone images before entering the fields and were provided special permission by farmers to enter only the 100 m$^2$ plots. The damage caused by entering a field to count rodent burrows and traps may be as large as the damage caused by the rodents. Using drones will hopefully allow farmers to determine the locations of the rodents without the need to enter the fields.

We aim to provide farmers the means to determine the location of rodents within their fields, allowing them only to add rodenticides in places with rodents, increasing the rodent control efficiency and decreasing the amount of rodenticides used. This, in turn, may allow farmers to change treatment protocols during the growing season according to within-field variations. For example, adding rodenticides or natural control (i.e., hunting perches [55]) only in small areas within fields, such as 100 m$^2$ used in this study, with lower NDVI and biomass, will not only decrease the amount and cost of rodenticides but should also increase the treatment effectiveness.

Previous studies conducted in Israel used rodent burrows as an index of Levant vole activity due to an inability to capture voles [21,45]. Still, this study is the first to determine the relationship between burrows and Levant vole numbers. A similar association was also found in European common voles [43]. Burrows can be used as an index of vole numbers, as it is much easier to count burrows than to trap voles; moreover, burrows can be calculated automatically using drones and machine learning, without the need to enter the fields [31].

Furthermore, we found significant positive relationships between biomass estimated from the drones and the crop yield harvested manually. Biomass calculated from drone images can be used as an estimate of yield instead of alfalfa harvested manually. The use of drones to calculate biomass can save time and can be applied during growing and before harvesting in areas where access is prohibited.

Currently, there are limitations when using drones. For example, in this study, we were limited by the height we could fly due to governmental regulations. Additionally, it took two days to use drones to take images of an entire field and to calculate the vegetation

indexes. Even though farmers cannot use traditional means of determining rodent locations, some may not want to invest this time and will prefer to add rodenticides. As UAVs cost will decrease and battery life, size, and speed will increase, as well as sensors' spatial resolution, so will the ability to use similar technologies at a much larger scale needed to monitor larger fields faster. Furthermore, as technology advances, it will be possible to cover larger areas using a swarm of UAVs [56]. If regulations change, flying higher will allow faster flight times and larger areas to be covered.

Regions in fields with multiple burrows and crop damage can be identified easily within time restrictions from drone images. There is a need to determine whether the use of both spectral vegetation indices (this study and [24]) and RGB drone imagery using machine learning [31] can increase the efficiency of determination of both rodent location and damage. There is also a pressing need to develop an innovative tool for farmers and researchers that will allow users to calculate both rodent presence and damage without physically entering the field, saving time and resources and increasing crop yield through precision rodent control. Furthermore, there is a need to develop automatic applications using drones that can dispense rodenticide and even flood fields [22] in areas with increased rodent activity, thereby increasing the efficiency of pest control, decreasing the amount of rodenticides, and saving water.

**Author Contributions:** D.K., I.I. and M.C. conceptualized the study; D.K. conducted the field research; D.K., A.B., D.M., I.I. and M.C. performed the analyses; D.K., A.B., D.M., I.I. and M.C. wrote the first draft. M.C. was responsible for funding acquisition. All authors have read and agreed to the published version of the manuscript.

**Funding:** Funding was provided by the Israel Ministry of Agriculture (#60-02-0003).

**Data Availability Statement:** Not applicable.

**Acknowledgments:** We thank Rafi Ettelberg, On Rabinovitz, Tito Nathanzon, Ezra Yassur, and the Agmon Hula Research Station for assistance in the field. We thank Keren Salinas from the Spectroscopy and Remote Sensing Laboratory (SRS) for assistance with image analyses. Trapping was conducted using a permit from the Israel Nature Parks Authority (Permit number: 2018/41815). This project was funded by the Ministry of Agriculture and Rural Development (# 60-02-0003).

**Conflicts of Interest:** The authors declare no conflict of interest.

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
