# Peer review of "The Use of Drones to Determine Rodent Location and Damage in Agricultural Crops"

_drones, doi:10.3390/drones6120396_

Round 1

Reviewer 1 Report (Previous Reviewer 4)

Please find my comments in the attached file. 

Author Response

I believe the reviewer 

Reviewer 2 Report (Previous Reviewer 3)

One major issue needs still to be addressed before publication. See below.

L146 write “Neural networks have been proposed…”

L205 You can safely round sensor width to two digits (12.83mm)

L205-209 reconsider rewriting sentence with better readability (split sentence?). Write “GSD of…”

L221 do you mean study sampling plot? Better rewrite as “Example image of a 10x10 sampling plot used in the study” or similar

L299/L311/L323/L346/L349 You answered me “Thank you for the comment. We indeed did use a linear mixed model but only provided the statistics. In the methods it is listed and we changed according to…”

That does not fulfil my request. The problem is not that you used a mixed effects model, which you do rightly because of the random effect “fields” in your case but that you do not take into account the nature of your response variable. The response value does not follow the Gaussian distribution because it is counts. It follows rather the Poisson distribution and should be treated differently in a regression model. One idea is to use a GENERALIZED linear mixed effects model (GLMM) with an appropriate Link function (e.g. log link) and family (e.g., PDF). These parameters should also be reported in the manuscript. The nature of your data set is not simple; I would highly suggest you consider a statistician with focus on test design and implementation to ensure the significance of your results reported in the paper.

Author Response

Reviewer 3 Report (New Reviewer)

This manuscript examines the potential use of a drone for measuring vegetation cover and the correlation to ground-based measures of rodent (mainly vole) densities, rodent burrow density, and crop yield (alfalfa).  The paper is of interest to those trying to better estimate rodent damage to crops in large fields.  One of the key areas that is missing here is reporting the efficiency of drone use.  Often in applications like this the effort (human labor and time) is often much higher than many ground-based techniques.  The authors need to review this in their manuscript by specifically stating the time investment needed to fly fields with drones, download the imagery, and analyze the imagery.  How long does that take?  Is it still a more efficient method than walking transects and counting burrows, or making ground-based estimates of crop cover and damage?  This is an important component of any such drone study and it currently is not in the manuscript but could easily (and should easily) be added by the authors. 

Abstract:

-I suggest adding a sentence or two about the efficiency of using drones (how many days and hours of flight time and image analysis it takes).

-Change to: “Burrow counting can therefore be used in place of trapping for voles…”  I think you must specify for voles because they are the ones that are making the burrows, right?  I think this is where we need more information about the setting—are voles and house mice always in these fields together and are there no other small mammals? I would think there is some evidence of competition where in high abundance of voles there are low abundance of house mice—is this correct? If it is mostly voles and they are the main ones making tunnels I think you need to keep it to them only for your predictions. 

Introduction:

-This is too long and there is far too much at the begging about agricultural threats to biodiversity.  This throws off the reader expecting to jump into the problem (i.e., identifying a more efficient sampling technique for these settings).  Consider starting the Introduction at line 60.

-Why are there two paragraphs in bold font in the Introduction?  This is distracting; please remove bold font.

-In the last paragraph of the Introduction, specify that NDVI will be measured by drone, and perhaps from <90 m altitude.  Your objective and hypotheses should be specific to your study and your study is on the adequacy of the use of drones as an improved method.

Methods:

-Again, why are some paragraphs bold font?  Please remove throughout the manuscript.

-Line 178:  Can you discuss why the traps were not baited?  Were the traps depressed beneath the vole tunnels in the grass?  How were the traps secured flat with the ground?  If the traps were only placed along vole runways then why are you catching anything but voles (eg, house mice)?  Was this the goal (only voles)?

-Line 188:  removed “the”

-Line 201:  this part of the sentence does not make sense.  Before entering?

-Line 205.  Why 20 m?  Can you base this on preliminary data or others before you that used this altitude successfully to accomplish a similar goal?  It seems the higher you fly the fewer flights (and flight time) would be required, but there must be a tradeoff with detection ability with your camera.

-Line 211:  We need more information about your flights and experimental setup.  Why 10 x 10 m plots?  These seem tiny.  How were those 10 x 10m plots chosen?  Did you stratify your design so they would get areas of dense and sparse vegetation cover?  Did you haphazardly choose plots in a field?  How close were plots to one another within a field?  Do you do all flights on the same day for a field?  Was the alfalfa all in the same stage among field (if so, what stage)?  Had farmers previously treated any of the fields for rodent management?  How long were flights?   Why 8-24 plots?  Were the fields with n = 8 plots those that were not very variable (ie, not patchy) compared to the fields with more plots?

-Line 213:  What is a subplot in this case?  The 10 x 10 m was stated as a plot, and the 1 x 10 m was ‘strips’.

-Line 225-261:  Is all this necessary?

-Figure 2:  There are 26 points here.  Interpretation is unclear.  What do the points represent?  When voles were zero, it would appear that you had 0,1,2,3,4,5,6,7 house mice in a plot.  Similarly for when house mice were zeros.  Can you give more of the data in the text, and also clarify this figure?

-Line 312:  give the exact p-value

-Line 322: You mean plot, not subplot, right?

-Line 337:  give the exact p-value

Discussion

-Delete the first paragraph and begin Discussion with the second.  We already got that info in the Introduction.

-In general, I feel the Discussion still needs work to better communicate the results and especially the interpretations.  See some of the points outlined above.  How scaleable is your study?  In other words, would you recommend that farmers take pictures of 10x10 m plots?  If so, how many, and how confident would you be in a very large area?  Would there be seasonal or crop periods that would be better or worse for this type of method?  In general, you should outline the limitations of your study.

Additionally, there needs to be more discussion about the relationship with voles and the other small mammals that were captured.  Is that an expected pattern to have high vole abundances when there are low house mice?  More needs to be discussed here and how that may influence your conclusions about this technology.  

The drone altitude seems to be an important issue to discuss more.  What are the limitations of the optical used and the altitudes that you could be confident identifying damage or burrowing?  If farmers wanted to do much larger areas, can you go higher and take less time because the ‘field of view’ of the optical is wider?

See initial comments in the Abstract for sharing the time it takes for personnel to use drones to adequately estimate damage or vole abundance.  Compare these hours to on-the-ground hours if you did not use drones.  Is it still a viable option, and in what types of situations?

Round 2

Reviewer 2 Report (Previous Reviewer 3)

The main concerns of the paper I had were addressed properly. The manuscript can be published in my opinion.

Author Response

Dear Reviewer

We thank you very much for your comments that greatly help us improve our paper

Thank you again!

Best

Motti

This manuscript is a resubmission of an earlier submission. The following is a list of the peer review reports and author responses from that submission.

Round 1

Reviewer 1 Report

The authors have presented a novel method to determine rodent location and damage in agricultural crops using UAS RGB photography. RGB is converted to NIR using a GAN-based method and the damage caused is estimated. The introduction is quite verbose, detailing the importance of the problem from an agricultural perspective. Some alternative methods to perform the same work are provided but not only conceptually compared. Related literature for the specific problem is scarce and should be enhanced.

The manuscript gives a very short description of the methodology used but not enough details are provided about the network used and configuration details. Some figures of the system and the results could improve the legibility of the document. The manuscript would benefit from a qualitative comparison between different methods.

Finally, in the disussion there is a part about results in Spain that is rather more suitable for the Results section.

Reviewer 2 Report

Dear Authors and Editors: 

The review report is an attached file in this email. 

Thank you. 

Sincerely yours, 

The reviewer

Reviewer 3 Report

The study investigates the relationship between rodents (burrows) and UAV imagery. Their investigation was conducted in nine alfalfa fields in Israel including 112 plots. Each plot (10x10m) was imaged with an UAV and an RGB camera in 20m altitude. The authors made some effort to estimate the NIR signal from the RGB camera in order to calculate the NDVI using machine learning. They conclude that NDVI is negatively correlated with Levant voles and that UAS imagery can be used to assess vole pressure.

The paper has some issues that needs to be solved before publication:

-        The statistics of the results is flawed. The burrow data is count data and should be treated as a Poisson distributed response variable. Thus, a generalized linear (mixed) model needs to be used in order to assess if the relationship to NDVI or biomass is significant.

-          It is unclear which model is shown in the scatter plots (Figure 2-5). This should be adequately described in the Figure captions and if needed in the material and methods section.

-          Model test data and the validation of the models should be better described in the paper. An extra section would be helpful for better understanding the overall evaluation process used in the study.

-          UAV imagery taken at 20m altitude has a very high ground resolution that should also outline rodent burrows. Discuss if not an object based approach (e.g., based on an object detection model) would provide much better relationships to rodent activity than the averaged NDVI approach used in this paper.

-          L138-145 check grammar and readability of this paragraph

-          L191 write GSD and provide accurate GSD value for the camera setting and altitude used in this study.

-          L280-285 This paragraph belongs in M&M section.

-          L307 This is a very strange section for me. An own section to provide the figures/tables? Figure and tables should be distributed in the appropriate sections they belong to and accurately referred and described in the surrounding text.

-           

Reviewer 4 Report

Please, find attached the file with my comments.

Reviewer 5 Report

I read both the manuscript and the answers to the previous reviewers' comments. In my opinion the manuscript is very well written and presents a sufficient contribution to be published in this journal.

Round 2

Reviewer 2 Report

Dear Authors and Editors:

Thank you for your revision.

Please find the suggestions for the second round below

 Thank you.

Sincerely yours,

The reviewer.

Reviewers Comments – Round 1

Author Response

Reviewer comments – Round 2

1. Introduction: the authors should present the study gap and what are the contributions of the authors in this study. They are essential to emphasize the specific findings in this study for both science and practice.

Thank you for the comment. We stated that the problem of monitoring rodents in agriculture. Here we when through what is currently known  and then go into how drones could in fact improve greatly the ability to both determine the presence of rodents and their damage for both applied and scientific studies.  

Thank you for your response.

- 2. Materials and Methods: typically, the two subsections of section 2 are: 21. Materials and 2.2 Methods, while the authors listed subsections 2.1 to subsection 2.7 do not specifically present the materials and methods. What is the specific law?

We followed the direction of the journal and re-checked. Please see the specific template provided by the journal here: https://www.mdpi.com/journal/drones/instructions

I think the authors did not understand my suggestion.

 Please point out which subsections are “Materials” and “Method.”

Please reconstruct the whole manuscript to make the entire manuscript more logical and coherent.

I am a native English speaker and furthermore a linguist read over the MS. We tried to improve the MS. If there is something specific could you be as so kind to tell us exactly what is not logical and coherent and we will be more than happy to fix it.

Please reconstruct as the reviewer suggested.

(above and below)

- Please remove subsection 3.6 Figures and Tables. This section is not essential and unnecessary.

We followed the direction of the journal and re-checked. Please see the specific template provided by the journal here: https://www.mdpi.com/journal/drones/instructions

I think the authors don’t understand my suggestion.

 Please delete this section and I believe it is unnecessary.

 You only need to add the Figures and the Tables close to the texts that you mention on them.

- Lines 351-358: please rewrite this sentence

We re-read the sentence and are unsure what to re-write. The below is correct English. Please see below. Could you be more specific please.

“Determining the location of rodents in agricultural crop fields is a very difficult task for farmers and researchers alike. For example, until now, decisions of where, when and how much rodenticides to add on a field were not based on accurate mapping of the locations of rodents in fields. Instead, farmers add rodenticides when they see signs of rodents in the fields; the effectiveness of this method is limited by the timing, size of the field, type of crop grown and preventative treatments used”

Determining the location of rodents in agricultural crop fields is a very difficult task for farmers and researchers alike. For example, until now, decisions of where, when and how much rodenticides to add on a field were not based on accurate mapping of the locations of rodents in fields.  Instead, farmers add rodenticides when they see signs of rodents in the fields; the effectiveness of this method is limited by the timing, size of the field, type of crop grown and preventative treatments used…”

(1)  In the topic sentence, the authors mentioned the difficulty in locating rodents. However, in the example, they discussed not only “the places where there are rodents” but also “the time when rodents come” and “how many rodenticides” to add to a field that is not related to the main topic.

This issue makes the example irrelevant and does not specify the vital idea the authors want to present(locations).

(2)  Weak adjective:

Words such as very, extremely, and totally are meant to intensify the adjectives you use, but they often have the opposite effect. For example, calling something very important suggests that the writer is not confident enough to simply declare that the thing is important and must instead rely on a crutch word. Try to reach for strong adjectives that can convey your meaning all by themselves.

Acceptable

That pizza was really tasty.

Better

That pizza was delicious.

Acceptable

Marie was very happy to be a part of the team.

Better

Marie was thrilled to be a part of the team.

Acceptable

The little boy was extremely hungry.

Better

The little boy was starving.

(3)  The word “fields” repeatedly appears in this text.

Consider using a synonym in its place.

There are only 84 words in this passage where 5 of the words ("field") are repeated.

And 50 words ("field") are repeated throughout the manuscript.

(4)  Possibly wordy sentence: Too many non-content words may indicate wordiness.

Consider rewriting to avoid some of these words: for, until now, of, where, when, and, how, much, to, on, a, were, the, in.

(5)  Is the preposition “on” after the verb “add” correct?

 It seems that the preposition used may be incorrect here.

(6)  Inconsistent spacing (before “Instead”):

It’s best to use consistent spacing between sentences within a document.

(7)   The parallel structure in this sentence having “the timing, size of the field, type of crop grown and preventive treatments”.

Please consider the singular or plural phrases of those.

(8)  The second sentence was used in the wrong tense.

Please consider the present perfect tense.

(9)  Wrong punctuation is this sentence: “Instead, farmers add rodenticides when they see signs of rodents in the fields; the effectiveness of this method is limited by the timing, size of the field, type of crop grown, and preventative treatments used (adding rodenticides at a specific point in the crop cycle without knowledge of rodent numbers)”.

This sentence confused the readers, and I think the readers cannot understand the authors’ ideas.

- References: To validate this study's results, your manuscript should build on the referenced  previous  method  (based  on  the  state-of-the-art  research  before  this)  to  compare the advancements of this research. Please consider removing [34], published in 1949, and they are too old and no longer suitable for current studies.

We thank you and deleted the reference 34 and left the more updated reference in the manuscript.

Thank you for revising.

- Please explain more specific applications of your studied drones when used in agriculture. For example, precisely sowing seeds and spraying pesticides or harvesting, etc.? So, what are the transport mass maximum/minimum of the drones in this study hypothesized? Will the speed of the drones affect their performance? Do the authors consider the safety of the drone when it moves in different terrain areas? How will the safety factor of drones in densely populated urban areas differ from the factor of safety in farm areas?

The application is this study is to determine the presence and damage of rodents using drone in agriculture.

For example, precisely sowing seeds and spraying pesticides or harvesting, etc.? So, what are the transport mass maximum/minimum of the drones in this study hypothesized?

 This is out of the context of the study. We did add, “Furthermore, there is a need to develop automatic applications using drones that can dispense rodenticide and even flood fields [60] in areas with increased rodent activity thereby increasing the efficiency of pest control, decrease the amount to rodenticides and save water. “

Will the speed of the drones affect their performance?

We used a constant speed.

 Do the authors consider the safety of the drone when it moves in different terrain areas?

This study was done only in alfalfa fields (this is a type of  a crop) and did not move between terrains.

How will the safety factor of drones in densely populated urban areas differ from the factor of safety in farm areas?

Here the drone was only flown in alfalfa fields and not in urban areas.

Thank you for responding.

I think the authors have not convinced me with these answers.

Please explain in detail not only for the reviewer but also for readers.

(in the texts of the manuscript, the technical information of drones that the authors use.)

The title: Please consider the suggestion as follows

Employing/Using unmanned aircraft systems (UAS) to determine rodents and damage in agriculture crops”

The reviewer cannot find the third affiliate in the manuscript. There are two second affiliates (

 Department of Geography and Environmental Studies, University of Haifa, 7 Mount Carmel, Haifa 3498838, Israel, Shamir Research Institute, Katzrin 1290000, Israel.)

Figure 2: Please be consistent when using the singular or plural form of " house mice/ house mouse" and "Levant voles/Levant vole".

Figure 5: the word "Biomass" overwrites some numbers in the Figure 5b.

Line 339 and Line 282: remove the dot after “Table 1” and lowercase the word “presents”